# Update on Dihydropteroate Synthase (DHPS) Mutations in *Pneumocystis jirovecii*

**DOI:** 10.3390/jof7100856

**Published:** 2021-10-13

**Authors:** Carmen de la Horra, Vicente Friaza, Rubén Morilla, Juan Delgado, Francisco J. Medrano, Robert F. Miller, Yaxsier de Armas, Enrique J. Calderón

**Affiliations:** 1Instituto de Biomedicina de Sevilla, Hospital Universitario Virgen del Rocío/Consejo Superior de Investigaciones Científicas/Universidad de Sevilla, 41013 Seville, Spain; cdlhorra-ibis@us.es (C.d.l.H.); rmorilla2@us.es (R.M.); juan.delgado.cuesta@gmail.com (J.D.); medrano@cica.es (F.J.M.); 2Centro de Investigación Biomédica en Red de Epidemiología y Salud Pública (CIBERESP), 28029 Madrid, Spain; 3Departamento de Enfermería, Universidad de Sevilla, 41009 Seville, Spain; 4Departamento de Medicina, Universidad de Sevilla, 41009 Seville, Spain; 5Institute for Global Health, University College London, London WC1E 6JB, UK; robert.miller@ucl.ac.uk; 6Faculty of Infectious and Tropical Diseases, London School of Hygiene and Tropical Medicine, London WC1E 7HT, UK; 7Department of Clinical Microbiology Diagnostic, Hospital Center of Institute of Tropical Medicine “Pedro Kourí”, Havana 11400, Cuba; Yaxsier@ipk.sld.cu; 8Pathology Department, Hospital Center of Institute of Tropical Medicine “Pedro Kourí,” Havana 11400, Cuba

**Keywords:** pneumocystis, dihydropteroate synthase, gene mutations

## Abstract

A *Pneumocystis jirovecii* is one of the most important microorganisms that cause pneumonia in immunosupressed individuals. The guideline for treatment and prophylaxis of Pneumocystis pneumonia (PcP) is the use of a combination of sulfa drug-containing trimethroprim and sulfamethoxazole. In the absence of a reliable method to culture Pneumocystis, molecular techniques have been developed to detect mutations in the dihydropteroate synthase gene, the target of sulfa drugs, where mutations are related to sulfa resistance in other microorganisms. The presence of dihydropteroate synthase (DHPS) mutations has been described at codon 55 and 57 and found almost around the world. In the current work, we analyzed the most common methods to identify these mutations, their geographical distribution around the world, and their clinical implications. In addition, we describe new emerging DHPS mutations. Other aspects, such as the possibility of transmitting *Pneumocystis* mutated organisms between susceptible patients is also described, as well as a brief summary of approaches to study these mutations in a heterologous expression system.

## 1. Introduction

*Pneumocystis jirovecii* is an atypical opportunistic fungus with lung tropism and worldwide distribution that continues to be one of the major opportunistic pathogens causing severe *Pneumocystis* pneumonia (PcP) in individuals with acquired immune deficiency syndrome (AIDS) and patients with immunosuppression due to other causes [1].

Dihydropteroate synthase (DHPS) is an essential enzyme in the metabolism of *P. jirovecii* involved in the synthesis of folic acid [2]. Co-trimoxazole, a combination of trimethoprim and sulfamethoxazole (TMP-SMX), is the drug of choice for chemoprophylaxis and therapy of PcP. Sulfamethoxazole inhibits the folate synthesis pathway through competition with para-aminobenzoic acid (pABA), one of the two substrates of DHPS. Widespread prophylaxis and treatment for *P. jirovecii* with sulfa drugs have decreased the incidence of PcP, but concerns have been raised about the possible emergence of *P. jirovecii* isolates resistant to these drugs. Point mutations in this gene have been associated with prior exposure to sulfa drugs in other microorganisms [3]. The most frequent mutations occur at the polymorphic positions 165 and 171, which lead to an amino acid change at positions 55 (Thr55Ala) and 57 (Pro57Ser) of the polypeptide chain, respectively. As is shown in Table 1, different strains with single and double amino acid substitutions at these positions have been identified [4]. Although chemoprophylaxis with sulfa-drugs is often associated with DHPS mutant infection, whether the presence of these polymorphisms is associated with poorer clinical outcomes remains controversial.

DHPS gene mutations were described for the first time in PcP patients with evidence of failure in chemoprophylaxis [5]. These mutations were present in 70% of AIDS patients receiving sulfa-drugs prophylaxis and were more common in patients from 1995 to 1997, although there was no evidence of sulfa drug resistance [4]. However, based on the homology with other microorganisms and studies used heterologous expression, mutations in the DHPS gene at positions 55 (Thr55Ala) and 57 (Pro57Ser) suggest the possibility that in *Pneumocystis* these polymorphisms might be responsible for some failures of sulfa-drugs prophylaxis in PcP patients [3,4,5].

## 2. Methods to Identify *DHPS* Gene Mutations

In the absence of a reliable culture method to study *Pneumocystis*, several techniques have been developed to improve the identification and analysis of DHPS mutations (Table 2) [6].

At first, the *DHPS* gene was identified by sequencing after specific PCR [7]. This method, still currently used, has been modified to improve the amplification in situations where there is a very low burden of the microorganism, as is the case for respiratory isolates obtained by non-invasive methods (sputum, oral wash, or nasopharyngeal aspirates). In these cases, to obtain higher quantities of DNA suitable for sequencing, other methods, such us touch-down PCR, have been developed [8].

Restriction fragment length polymorphism (RFLP) assay is another of the main techniques developed to identify mutations at 55 and 57 codons of the *Pneumocystis DHPS* gene. The RFLP assay was compared with direct DNA sequencing on PcP isolates describing in all the samples the RFLP assay correctly the DHPS polymorphisms [9]. The advantages of this method are that it can be performed within one day and permits a fast, cost-effective, and reliable method of detecting DHPS mutations [9,10,11]. For this reason, it has been used worldwide mainly in PcP patients but also in individuals colonized by *P. jirovecii* (PjC) [10,11,12,13,14,15].

Another extensively used method to identify the *DHPS* gene was the single-strand conformation polymorphism (SSCP) that allows rapid detection of these mutations [16]. Using plasmid constructs as control, SSCP could detect mutations in as little as 10% of a minority population of DNA. The comparison between SSCP and PCR sequencing showed consistent results for all isolates except that in some of the samples, mutations were detected by SSCP but not by direct sequencing. To confirm the real result, sequencing of individual clones after subcloning confirmed the presence of mutations as was determined by SSCP, concluding that this was a very simple and sensitive method for rapid identification of DHPS mutations [16]. Several groups have used this technique until recently [17,18,19,20,21].

However, in recent years, real-time PCR methods have been developed that are able to genotype more samples than in the previously described DHPS-PCR followed by direct sequencing [22]. Real-time PCR is a suitable method for PcP diagnosis and detection of DHPS mutants, but no data are available about its use in PjC patients.

All these techniques have advantages and disadvantages. In particular, the sensitivity of the methods to multiple genotype detection in a given sample differs among them. Sanger sequencing-based methods are less sensitive than next-generation sequencing (NGS) technologies because minor variant detection is generally restricted to strains with a relative abundance of DNA [23]. This must be kept in mind for interpreting results of studies.

Furthermore, as added value, identification of DHPS polymorphisms is useful for epidemiological studies that lead to describing the route of transmission or sources of infection. For this reason, it has been included in multi locus epidemiological studies, such as the one described by Esteves and colleagues [24]. In this study, the high-throughput typing strategy for *P. jirovecii* characterization was performed from DNA pooling by quantitative real-time PCR, followed by multiplex-PCR/single base extension describing the most relevant polymorphisms at four loci (mt85, SOD110, SOD215, DHFR312, DHPS165, and DHPS171) [24]. This strategy allows identification of a large number of samples. Therefore, multiplex-PCR/single base extension could be a powerful tool to perform overall epidemiological studies of *P. jirovecii* including DHPS gene polymorphisms. However, DHPS is a nuclear single-copy gene in *P. jirovecii* and consequently more difficult to amplify than mitochondrial multicopy genes [8,12].

## 3. Geographic Distribution

The prevalence of DHPS mutations has been extensively studied in developed countries where it is significantly higher than in developing countries and could be associated with prior exposure to sulfa drugs. Studies conducted in the USA demonstrate a higher prevalence of *P. jirovecii* mutations, in comparison with European countries, and even lower rates have been reported in developing countries [4,5,13,14,15,18].

### 3.1. Developing Countries

Recent reports have described increases of PcP cases in Africa, Latin-America, and Asia, but a limited number of studies have evaluated the presence of *Pneumocystis* DHPS mutations in developing countries [25]. In these areas, the majority of studies report a lower prevalence of DHPS mutations.

In African countries, several studies (up to 2005) have described low frequencies of DHPS mutations. The overall rate of DHPS gene mutations was 7.1% among AIDS-related PcP patients in Harare, Zimbabwe [26]. In South Africa, the rate of DHPS mutations varied from 1.3% in adults with PcP up to 13.3% in HIV-infected children [27,28]. This low frequency was related to the lack of exposure to TMP-SMX in the populations studied during those periods. However, a few years later, in the same geographical area in South Africa, a prevalence of 56% of DHPS mutations was reported in adult AIDS-related PcP patients [29]. This high frequency of DHPS mutations was also recently described in immunosuppressed children from Mozambique with a rate of DHPS mutations of 24.9% in codon 55 (A165G) and 52.8% in codon 57 (C171T) [24]. Furthermore, an epidemiological study performed on PcP patients from Uganda showed that almost 100% of patients evaluated had mutations at codon 57 [30]. With regard to the North African population, a single study, conducted in Tunisia reported an 11% rate of DHPS mutations [31]. This evidence could suggest that variations in DHPS mutations rates may be due to differences in time periods analyzed, populations analyzed, or study designs.

Several studies conducted in Latin American analyzed the prevalence and distribution of *P. jirovecii* including DHPS gene mutations. As is shown in Table 3, low frequencies of mutated strains were described in Latin America in Brazilian, Colombian, and Cuban populations.

In Brazil, a pioneering study in PcP patients showed an absence of DHPS gene mutations. In contrast, a recent report described clusters of *P. jirovecii* with DHPS mutations among immunosuppressed patients following renal transplantation [32,33]. Among Colombian patients with PcP or PjC, DHPS mutations in respiratory samples were detected at a low rate [34].

The first study conducted in Cuba in young children with whooping cough showed a frequency of 18% mutated strains [35]. These data are like that from a more recent study that determined DHPS gene mutations using MLST in a pool of samples from Cuban children where rates of 13.3% for A165G and 4.8% for C171T were reported [24]. Detection of low rates of DHPS mutations among Cuban children could be due to a lack of use of TMP-SMX in this population.

In Chile, high frequencies of DHPS mutants have recently been described [13]. *P. jirovecii* with DHPS mutations was present in over of 40% of PcP patients [13]. The frequency of DHPS mutations among immunosuppressed patients from French Guiana was 14.2% [21].

In Asia, studies carried out during the last decade reveal that the presence of *P. jirovecii* harboring DHPS mutations was low or absent. Most of the studies used PCR RFLP on immunosuppressed PcP patients and showed an absence of DHPS mutations in Turkey, Korea, China, and India [36,37,38,39]. Interestingly, in India, mutations rates between 4.1% and 6.2% have been described in other studies [40,41] as well as mutations at codon 171 without mutations at the commonest positions of codon 55 and 57 [42]. Recently, the presence of DHPS mutation at codon 55 was recently observed in one out of 60 AIDS-related PcP patients in China [43], and a prevalence of 1.5% of DHPS mutations has been reported among hospitalized patients from Turkey [44].

In Iran, an overall rate of DHPS mutations of 14.7% was recently described [14]. These percentages are like data reported from Thailand, where a rate of 11.7% was described in PcP patients [45].

Several considerations could be established in the analysis of mutations rates in the DHPS gene in developing countries. On the one hand, the presence of of *P. jirovecii* with mutations in the DHPS gene could be associated with sulfa drug use in the treatment or prophylaxis of PcP, but on the other hand, the increased widespread use of TMP-SMX for other infections could induce the appearance of mutations and *P. jirovecii* containing DHPS mutations can then subsequently be transmitted. By contrast, the limited use of sulfa drugs may also favor the absence of DHPS mutations in the *P. jirovecii DHPS* gene, as has been described in the few studies available from Asian and Latin American countries [25].

### 3.2. Developed Countries

In developed countries, the epidemiology and clinical features of PcP have been extensively defined and well documented (Table 4).

In the USA, since 1998, many studies have shown the presence of DHPS mutations in PcP cases in rates from 35% to 81% [5,46,47,48]. High overall frequencies of DHPS mutations in different periods from 1976 to 2001 have been described, with the rates of DHPS mutations being lower before the 1990s and the highest in San Francisco at the end of the 1990s [46]. In this city, several reports consistently described overall high frequencies of mutations in the *DHPS* gene (81.5% (1996–1999), 81.4% (1997–2002)) [48,49]. A recent multicenter report compared the rate of DHPS mutations in several areas of the world (including USA) between 2010 and 2015, finding mutation rates of 44.6% (A165G) and 50% (C171T) [24].

In Europe, several studies, most of them performed on patients with PcP, have shown differences in the prevalence of DHPS mutations. The presence of strains harboring mutations varies between 0% and 50%, depending on the patients studied, period assayed, and/or geographic localization and most of them have contributed to drawing a map of DHPS gene mutations. France showed a lower rate of DHPS mutations than other European countries, with the highest rate of mutations being described in Paris and Lyon among patients with PcP [10,19,50], and was equally distributed among several groups of patients [51]. However, in more recent French studies, the prevalence of *P. jirovecii* with DHPS mutations is lower or even absent [52,53,54]. A decreased frequency of DHPS mutations in other countries including Switzerland, Italy, and Portugal has also been observed [15,18,20,22,55,56,57,58]. Recent studies of DHPS mutations in North and Central Europe have also shown a low prevalence of DHPS mutations related to the scarce use of sulfa drugs [21,59,60,61,62,63]. Only one study from Denmark of patients with PcP from before 2000 reported a prevalence of DHPS mutants of 20.4% [64]. In Spain, several groups have described the presence of DHPS mutations in patients with PcP and PjC. The evolution in the rates of mutations varied from 40% to 10% in PcP patients, and in PjC patients decreased from 22% to 7.5% [12,15,22,65,66,67]. In this line, a single-center study performed in Spain evaluated HIV-infected patients with PcP from 1998 to 2010 and showed that *P. jirovecii* harboring DHPS mutations decreased from 33% before HAART was available to 5.5% post HAART [68]. Similarly, in the UK, among HIV-infected patients with PcP, a rate of 36% of mutations was described between 1992–1993 and had reduced to 16.6% seven years later [26]. Other studies from other geographical areas confirm this downward trend in rates of DHPS mutations. In Japan, rates changed from 25% in PcP patients exposed and non-exposed to sulfa drugs in the early 2000s to be very low in a most recent report [69,70,71]. However, the numbers in those studies are small and the data must be taken with caution. In Australia, a single report described a 13% prevalence of DHPS gene mutations in isolates from patients with PcP [72].

## 4. Mutation Effect on Treatment Outcome

The effect of DHPS mutations on treatment outcome remains controversial. Studies that have assessed this issue offer conflicting results. The mortality of AIDS-related PcP seems to be more related to the initial clinical severity of the pneumonia rather than to the presence of DHPS mutations, which seem more related to a loss of efficacy of TMP SMX as prophylaxis [50,64,70,73].

### 4.1. Impact of DHPS Mutations on PcP Outcome

In one study of HIV-infected patients with PcP conducted in the USA, the majority of patients with DPHS mutations survived, and those with mutant genotypes were more likely to require mechanical ventilation and intensive care within 72 h hospitalization and tended to have worse outcomes compared with those who did not have DHPS mutations [49]. In addition, an association between DHPS mutations and greater severity of PcP defined by higher LDH, need for intubation, and death has been observed [55].

However, a systematic review based on 13 studies concluded that this association was weaker in studies performed after 1996 but stronger in studies that included multiple isolates per patient. It confirmed that the risk of developing DHPS mutations was higher in PcP patients who had received sulfa prophylaxis. In this meta-analysis, the effects of DHPS mutations on clinical outcome were inconsistent regarding the presence or absence of an association [74].

In spite of these inconsistencies, many studies have suggested that mutations in DHPS could be associated with a poor prognosis in PcP, and mutations may develop it as a result of exposure to sulfa-drugs. Regarding the outcome of PcP patients, a study based on multivariate analysis revealed that 3-month survival was lower in patients who harbored DHPS mutations [64]. Additionally, DHPS mutations were more common in patients who had previous exposure to sulfa-drugs. Whether the increased mortality was due to failure of TMP-SMX treatment for PcP is unclear, as most patients with DHPS gene mutations responded to PcP therapy with sulfa drugs [64]. In a multicenter study, the majority of AIDS-related PcP patients with DHPS mutations respond to sulfa or sulfone therapy, although the response rate to these agents was lower than the rate among those who had a wild-type strain [75]. This data agreed with results obtained in PcP patients from Japan, suggesting that treatment failed more frequently in patients whose isolates had DHPS mutations than in those which harbored wild-type strains [70].

In 2001, a multicenter prospective study analyzed the effect of DHPS mutations on outcome of PcP in HIV patients and, in contrast with other reports, showed that those PcP patients with wild-type *Pneumocystis* did not have better outcomes than patients with DHPS mutant strains when both were treated with sulfa drugs [73]. It was concluded that the presence of DHPS mutants should be only one of several criteria guiding the selection of initial drug treatment in PcP patients. Similar results were obtained in PcP patients from European countries, such as Portugal, Italy, France, Switzerland, and Spain [18,66,76,77,78].

In 2008, an epidemiological study performed in Spain showed a lower rate of DHPS mutations in PcP patients who did not have a worse outcome [79]. More recently, a larger study concluded that DHPS mutations did not relate to a higher mortality [68]. Similar to this, another study from Japan did not find a relationship between mutations and a poor prognosis [71]. These results contradict data from Australia, where patients with DHPS mutations were more likely to progress to severe PcP, with a need for invasive ventilation and ultimately having a poor outcome [72].

A large epidemiological study concluded that mutations were more frequent in the USA than in Europe, but analysis of clinical data showed that the mortality related to the underlying severity of the illness [80]. More recently, a study of 301 patients with PcP showed sulfa prophylaxis increased the risk of infection with pure mutant but not mixed genotypes, but once again, the presence of DHPS mutations was not associated with increased mortality; however, the authors suggest further studies are needed to clarify the role of each DHPS mutation [81].

A recent study from Chile reported that PcP patients with mutant genotypes were more likely to suffer sulfa treatment-limiting adverse reactions and to have a twice-longer duration of mechanical ventilation, suggesting a decreased efficacy of TMP-SMX [13]. Similarly, a large number of PcP cases with DHPS mutations were described in Uganda, and interestingly mortality was related to mutation at 57 codon [30].

On the other hand, recent studies have identified new DHPS mutations at nucleotide position 288 (Val96Ile) and 294 (Glu98Gln) that appear to be associated with treatment failure and mortality in pediatric HIV-infected and uninfected and adult HIV-infected patients with PcP [82,83]. However, further research is needed to improve our understanding of the significance of these recently described mutations in the DHPS gene.

### 4.2. Impact of DHPS Mutations on PcP Chemoprophylaxis

Duration of sulfa or sulfone chemoprophylaxis increased the risk of developing *Pneumocystis* DHPS mutations [75]. Several studies have consistently demonstrated a significant association between the presence of DHPS gene mutations and failure of sulfa prophylaxis [4,5,16]. Therefore, physicians should be alert to the risk for drug resistance during sulfa prophylaxis because the detection of DHPS mutations may allow a change in chemoprophylaxis, reducing the risk of PcP [67].

## 5. Transmission

DHPS mutations are frequently associated with the use of prophylaxis with sulfa drugs [4,5]. A study of relapsed PcP in AIDS patients suggests that DHPS mutations can be selected de novo within patients by the pressure of a sulfa or sulfone drug and that DHPS mutations may be associated with reactivation of a silent *P. jirovecii* infection [19]. However, other factors could contribute to the presence of *P. jirovecii* harboring DHPS mutations. Person-to-person transmission of mutated strains from individuals in whom de novo mutation is occurring, or who have previously been exposed to sulfa drugs, to patients without prior sulfa-prophylaxis may explain the high proportion of mutations observed [19]. Several studies conducted in order to clarify this issue have been reported. A study performed on PcP patients found a mutation at amino acid 57 in all patients receiving sulfa-drugs prophylaxis but only in 20% of patients who did not receive prophylaxis [81]. Another study of PcP patients from Portugal showed that DHPS gene mutations were frequent in the immunosuppressed population but were not associated with previous sulfa exposure [77]. Both studies suggested a possible and accidental acquisition and transmission of *P. jirovecii* with mutant DHPS genotypes.

During the last decade, several studies were carried out to identify DHPS gene mutations in patients with a variety of underlying diseases and without previous exposure to sulfa drugs [12,51,84]. Totet et al. described the presence of similar patterns of DHPS mutants among different populations, such as PcP patients, PjC and non-immunosuppressed infants, suggesting that presence of mutations was independent of the underlying condition or previous receipt of sulfa prophylaxis [51]. These data are supported by the results obtained contemporaneously in Spain and are in agreement with other studies in which the DHPS mutations were identified in individuals with PjC and without previous sulfa exposure, such as patients with chronic bronchial diseases or infants and toddlers with whooping cough [35,67,84]. The actual transmission of mutated *P jirovecii* strains had been described in a cluster of PcP [85]. In addition, the role of colonized patients as a source of *P. jirovecii* infection has been described [86]. Taken together, these studies suggest that colonized patients could play a role in the maintenance and transmission of the *P. jirovecii* with mutations in the DHPS gene and may represent a source of infection for susceptible individuals to develop PcP.

A prospective epidemiological study of PcP patients from three European cities showed nosocomial transmission of *P. jirovecii* that had mutations in the *DHPS* gene that could explain the high proportion of mutations in patients without exposure to sulfa drugs [19]. Additionally, possible nosocomial transmission has been investigated in other locations [33,87,88]. In France, longitudinal screening of patients with PcP and PjC suggested that circulation of *Pneumocystis* exists within hospitals [87], and more recently, the presence of *Pneumocystis* harboring DHPS mutant genotypes in the environment of a bronchoscopy unit has been described in Spain [88].

Taking into account all of these studies of *Pneumocystis* transmission, there is consistent evidence that *P. jirovecii* with DHPS mutations could be transmitted from person to person and could represent a public health risk, especially in the hospital environment. In order to improve prevention and control of this infection in a nosocomial setting, further studies will be necessary.

## 6. Models for Studying Pneumocystis DHPS Mutations Using Other Microorganisms

As has been already described, the most frequent polymorphisms are two mutations leading to two amino acid changes (Trp-55-Ala and Arg-57-Ser), observed as a single or double mutation in the same *P. jirovecii* isolate. Knowing whether mutations present in the *P. jirovecii DHPS* gene leads to resistance to sulfa drugs is hampered by the lack of an in vitro culture system for *P. jirovecii*. In the absence of a long-term culture system for *P. jirovecii*, the resistance conferred by these mutations cannot be directly assessed and consequently several models in other microorganisms have had to be developed to address this issue.

### 6.1. Saccharomyces cerevisiae

The potential resistance to sulfa drugs conferred by changes in these amino acids was investigated using an *S. cerevisiae* model [89]. Single or double mutations identical to those observed in the DHPS gene from *P. jirovecii* were introduced by in vitro site-directed mutagenesis into alleles of the *S. cerevisiae FOL1* gene. Mutated alleles were integrated at genomic locus in *S. cerevisiae* and expressed by functional complementation in a strain with a previously suppressed *FOL1* gene. The data obtained from this model showed that mutation at codon 55 conferred resistance to sulfanilamide whereas mutation at codon 57 conferred resistance to sulfanilamide and sulfadoxine. The results related to sulfadoxine were consistent with epidemiological data from *P. jirovecii* that show an association between DHPS mutation and failure of pyrimethamine/sulfadoxine prophylaxis in immunosuppressed patients. [19]. Both mutations separately conferred hypersensitivity to sulfamethoxazole and dapsone. In addition, double mutation conferred hypersensitivity to sulfamethoxazole in *S. cerevisiae*, which contrasts with epidemiological data about *P. jirovecii*.

Illiades and coworkers developed a model where DHPS mutations commonly observed in *P. jirovecii* were reverse engineered into the DHPS gene of *S. cerevisiae* [90]. Those mutations, T(597)A and P(599)S, that may occur separately but are commonly found together and are associated with SMX treatment failure. Double mutants showed resistance to and were able grow in the presence of sulfa drugs, which reveals increased resistance to sulfa drugs [90].

More recently, another *S. cerevisiae* model was used to examine the amplified *P. jirovecii DHPS* gene from clinical specimens, cloning this amplicon into a centromeric plasmid to transfect DHPS-deleted yeast strains that allows for a fully effective complementation [91]. Yeast cloned with the double mutation showed reduced sensitivity to SMX compared to strains complemented with single mutations or strains with the wild type. The data from this study suggests that this *S. cerevisiae* model could be useful for studying the biological properties of *P. jirovecii*.

### 6.2. Escherichia coli

Another approach to investigate drug treatment failure and sulfa drug resistance was evaluated in *E. coli* [92,93]. The *P. jirovecii* genomic sequence encoding FAS, a trifunctional protein that includes DHPS was identified by PCR amplification from BAL samples obtained from PcP patients. The *P. jirovecii* trifunctional genes (*PjFAs*) were cloned and expressed in a DHPS-disrupted *E. coli* strain [94]. This model allowed testing of the role of DHPS mutations. It established that the presence of substitutions T(597)A and P(599)S in the DHPS domain of *PjFAs* led to cross-resistance against sulfa drugs and the presence of both mutations led to increased sulfa drug resistance whatever has been tested on other pathogens than *Pneumocystis* as *Saccharomyces* or *Escherichia* [93]. The evaluation of sulfa drugs has shown that mutation in codon 55 (T597A) leads to increased sulfa drug sensitivity, whereas the single amino acid substitution at codon 57 (P599S) leads to sulfa drug resistance. As already described, the presence of a double mutation was cooperative and increased sulfa drug resistance. In the other hand, a new double mutant (T597V P599S) showed a higher sulfa drug resistance [94].

### 6.3. Mathematical Model

Recently, a quantitative model to predict changes in the binding affinity of inhibitors of the mutated proteins has been developed [95]. Predicted changes in binding affinity upon mutations were highly correlated with experimentally measured data. The model showed similar or better performance when compared with the resistance data for the drug/target pair (Pj DHPS/SMX). Potentially, this mathematical prediction model could be useful in the evaluation of possible resistance of known and newly identified mutations in the nucleotide sequence.

## 7. Conclusions

Taken together, it can be concluded that DHPS mutations in *P. jirovecii* are widely spread around the world. The different geographical prevalence among mutations of the DHPS gene is not only related to use of sulfa prophylaxis, but also in the person-to-person transmission, which could also contribute to spreading *P. jirovecii* that harbors DHPS mutations.

It can be assumed that TMP-SMX prophylaxis induces the appearance of DHPS gene mutations. These DHPS mutations have been associated with resistance to sulfa drugs using in vitro assays. However, the presence of mutations in PcP patients and resistance to treatment with sulfa drugs remains unclear. Some studies suggest that DHPS mutations are associated with a worse outcome (defined as a need for mechanical ventilation and higher mortality). However, large prospective studies are needed to clarify the role of DHPS mutations, especially the recently described mutations, and outcome for patients with PcP.

Finally, individuals colonized with *P. jirovecii* with DHPS mutations could play a role in the transmission of these mutations, representing a reservoir and source of infection especially in certain areas, such as the hospital environment. Further research is needed to improve our understanding of significances of *P. jirovecii* thar boring DHPS gene mutations.

## Figures and Tables

**Table 1 jof-07-00856-t001:** Commonly reported amino acid substitutions observed in the DHPS gene of *Pneumocystis jirovecii*.

DHPS Genotype Name	Amino Acid at Position 55	Amino Acid at Position 57
Wild Type		
Genotype 1	Threonine	Proline
Mutant		
Genotype 2	Alanine instead of Threonine	Proline
Genotype 3	Threonine	Serine instead of Proline
Genotype 4	Alanine instead of Threonine	Serine instead of Proline

**Table 2 jof-07-00856-t002:** Characteristics of different typing methods used for genotyping of DHPS in *Pneumocystis jirovecii*.

Method	Reproducibility	Portability	Discriminator and Power	Ease of Use	Ease of Interpretation	Setup Cost	Cost/Isolate	Time to Result (Days)
RFLP	High	Poor	Moderate	Easy	Moderate	Moderate	Low	1
PCR seq	Very high	Good	High	Moderate	Moderate	Moderate	Moderate	1
SSCP	High	Poor	Moderate	Moderate	Moderate	Moderate	Moderate	2
MLST	Very High	Excellent	Moderate	Moderate	Moderate	High	High	2

Adapted of reference [6]. RFLP: restriction fragment length polymorphism; PCR seq: PCR/nested PCR and sequence specific gene; SSCP: Single-Stranded Conformational Polymorphism; MLST: multilocus sequence typing.

**Table 3 jof-07-00856-t003:** Distribution of DHPS mutated strains in developing countries.

Continent	Country (Area)	Molecular Evidence %	DHPS Detection	Samples/Immune Status	Period of Study	Author, Year
Africa	Mozambique	24.9% codon 55/ 52.8% codon 57	MLST	22 NP IS children	2006–2008	Esteves F, 2016
	Uganda	100%	PCR seq	13 AIDS-PCP	2007–2009	Taylor SM, 2012
	South Africa	56%	PCR seq	151 PCP	2006–2007	Dini L. 2010
		1.8%	PCR seq	53 PCP	2000–2003	Robberts, 2005
		13.3%	PCR seq	30 AIDS PCP children	No available	Zar HJ, 2004
	Zimbabwe	7.1%	PCR seq	14 AIDS-PCP	1992–1993	Miller RF, 2003
	Tunisia	11.7%	RFLP	17 PCP	2005–2008	Jarboui MA, 2011
Latin America	Brazil	No mutations	PCR seq	PCP Fixed tissues	1997–2004	Wissmann G, 2006
	Chile	48% (HIV) 50% (nHIV)	RFLP	56 PCP HIV and nHIV	2002–2010	Ponce C, 2017
	Colombia	7.7%	PCR seq	88 PCP and PjC	2004–2005	Muñoz C, 2012
	Cuba	13.3% codon 55 and 4.8% codon 57 18%	MLST RFLP	10 IC children 163 IC children	2013 2010–2013	Esteves F, 2016 Monroy Vaca E, 2014
	France Guiana	14.2% mutations	MLST/RFLP	7 AIDS/7 IS	2012–2014	Le Gal S. 2015.
Asia	Turkey	1.5% (One with acute myeloid leukemia and one with lung cancer)	RFLP	137 both IC and IS	2016–2019	Gurbuz CE, 2021
	Thailand	11.7%	RFLP	17 AIDS-PCP	1997–2003	Siripattanapipong S, 2008
	China	1.6% 12% 7%	RFLP PCR seq PCR seq	60 AIDS-PCP 24 AIDS-PCP 15 AIDS-PCP	2015–2016 2009–2013 1998–2001	Zhu M, 2020 Deng X, 2014 Kazanjian PH, 2004
	India	No mutations 55 or 57 4.1% 6.2% No mutations detected	PCR seq PCR seq PCR seq RFLP	14 AIDS-PCP 24 PCP nHIV 16 AIDS-PCP 4 PCP	No available 2006–2009 2006–2009 No available	Mane A, 2015 Tyagi AK, 2011 Tyagi AK, 2010 Tyagi AK, 2008
	Iran	14.7% (6.25 HIV, 27.3% IS and 14.3% COPD)	RFLP	34 PCP/PjC	2010–2011	Sheikholeslami MF, 2015

PcP: *Pneumocystis* Pneumonia; PjC: *Pneumocystis* Colonized; IS: Immunosupressed; IC: Immunocompetent; NP: nasopharyngeal swabs; nHIV: non HIV; COPD: chronic obstructive pulmonary disease; RFLP: restriction fragment length polymorphism; PCR seq: PCR/nested PCR and sequence specific gene; SSCP: Single-Stranded Conformational Polymorphism; MLST: multilocus sequence typing.

**Table 4 jof-07-00856-t004:** Distribution of *P. jirovecii* with DHPS mutations in developed countries.

Developed Countries	Area	Molecular Evidence	DHPS Detection	Population	Period of Study	Author, Year
USA	3 cities	0% nHIV/35% AIDS	PCR seq	20 HIV and 7 nHIV with PCP	1976–1997	Kazanjian P,1998
	5 cities 3 cities San Francisco	69.1% 68.5% 81.4%	PCR seq PCR seq PCR seq	191 AIDS-PCP 111 AIDS-PCP 197 AIDS-PCP	1995–1998 1996–1999 1997–2002	Beard CB, 2000 Huang L, 2000 Crothers K, 2005
	San Francisco	44.6% codon 55 and 50% 57	MLST	30 AIDS-PCP	2004–2012	Esteves F, 2016
France	Reims	6.2%	RFLP	48 PCP and PjC	2008–2013	Nevez G, 2018
	Rennes	Absence	RFLP	84 PCP	2008–2011	Le Gal S, 2013
	Lyon	33% DHPS mutations	SSCP	112 AIDS-PCP	1993–1996	Rabodonirina M, 2013
	Paris	18.5 %, 12.5% mixed strains	RFLP	993 PCP (65% HIV)	2003–2008	Magne D, 2011
		17.4%	RFLP	BAL/PCP	1998–2001	Latouche S, 2003
	Brest	0% from Brest (2 patients with mutations from Paris)	RFLP	63 IS and IC	2007–2010	Le Gal S, 2013
	France	Presence in PCP, PjC, infants	RFLP	13 PCP-8 PjC-18 IF	1996–2001	Totet A, 2004
North/ Middle Europe	Germany (Cologne)	2.8%	PCR seq	PCP 35 HIV/60 nHIV	2011–2016	Suarez I, 2017
	Poland	8% at 55 position	RFLP	15 HIV/10 nHIV	No available	Golab E, 2007
	Sweden	Absence mutations	PCR-Pyrosequencing	103 IS	1996–2003	Beser J, 2012
	Netherland (Maastricht)	13.7%	RFLP	595 IS and IC	1997–2010	Vanspauwen M, 2014
	Denmark	20.4%	PCR Seq	144 PCP-AIDS	1989–1999	Helweg-Larsen J, 1999
	Belgium (Brussels)	4.5%	PCR Seq	120 (PCP and PjC)	2014–2015	Montesinos I, 2017
Portugal	Lisbon	8% 55–14% 57	MLST	182 (IC)	2010–2015	Esteves F, 2016
	Lisbon	8% 8.3% 27% 46%	RFLP RFLP RFLP RFLP	52 HIV and nHIV 12 PCP PCP HIV and nHIV 42 PCP	2001–2004 2002–2003 1994–2001 1994–1997	Esteves F, 2008 Costa MC, 2003 Costa MC, 2003 Matos O, 2003
Switzerland	Lausanne Zurich	7.4% 10%	SSCP SSCP	131 PCP 54 PCP	1990–2000 1991–1998	Hauser PM, 2010 Hauser PM, 2010
Spain	Seville	10% 55 and 4.8% 57	MLST	7 HIV IS and 5 COPD	2006–2013	Esteves F, 2016
	Seville	40 PCP and 22 PjC	RFLP	25 PCP, 50 PjC	2005–2010	Friaza V, 2010
	Barcelona Several cities Seville Seville Seville	33% preHAART and 5.5% post 3.7% 29% 35.5% 25% on COPD /33% on AIDS	PCR Seq PCR Seq RFLP RFLP RFLP	197 AIDS-PCP 207 AIDS-PCP 53 PCP and PjC 15 PCP, 238 PjC 15 COPD and 7 AIDS-PCP	1989–2004 2000–2004 2001–2004 2001–2003 2001–2002	Alvarez Martinez M,2010 Alvarez Martinez M,2008 Esteves F, 2008 Montes Cano M,2004 Calderon E, 2004
Italy	Milan Milan Milan	9.1% 7.6% 8.4%	RFLP RFLP SSCP	207 AIDS-PCP 261 AIDS-PCP 107 AIDS-PCP	1994–2004 1996–2006 1994–2001	Valerio A, 2007 Valerio A, 2006 Ma L, 2002
UK	London London	36% 16.6%	PCR Seq PCR Seq	25 AIDS-PCP 12 AIDS-PCP	1992–1993 2000–2001	Miller RF, 2003 Miller RF, 2003
Australia	Sydney	13%	PCR Seq	68 PCP HIV and nHIV	2001–2007	Van Hal SJ, 2009
Japan	Tokyo Tokyo	26.5% 25%	PCR Seq PCR Seq	34 PCP 24 PCP HIV and nHIV	1994–1999	Takahashi T, 2002 Takahashi T, 2000
Korea	Seoul	Absence mutations	PCR seq	95 PCP	2007–2013	Kim T, 2015

PcP: *Pneumocystis* Pneumonia; PjC: *Pneumocystis* Colonized; IS: Immunosupressed; IC: Immunocompetent; nHIV: non HIV; RFLP: restriction fragment length polymorphism; PCR seq: PCR/nested PCR and sequence specific gene; SSCP: Single-Stranded Conformational Polymorphism; MLST: multilocus sequence typing.

## Data Availability

Not applicable.

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
