# Peer review of "Update on Dihydropteroate Synthase (DHPS) Mutations in Pneumocystis jirovecii"

_jof, 2021, doi:10.3390/jof7100856_

Round 1

Reviewer 1 Report

Thank you for this review focusing on the epidemiology of DHPS mutations in Pneumocystis jirovecii and on their impact on PCP treatment. It gathers a lot of data on this interesting and controversial subject. To my mind, although the different parts are well organized, the structure of the paragraphs composing these parts (especially part 3 and 4) can be a little confusing for it gives a lot of information not necessarly in a logical order (part 4) or gives too many information (part 3).

Major comments

  • Part 2:

Please consider adding some lines on the possibility of mixed infections. It has been reported in several studies, especially using NGS and only evoked lines 86-91 in this review. DOI: 10.3201/eid2308.161295 , DOI: 10.1371/journal.pone.0125763

  • Part 3:

Please consider lightening this section of the text, especially as all numbers are already detailed in tables 3 and 4. (discordance between line 199 and study of Beard CB,2000?)

The relationship between mutations and use of TMP-SMX as prophylaxis is repeated several times in the part 3. Please consider adding references on the use of prophylaxis in the particular cited situations; Or evoking it as a global suggestion concerning the different rate of mutations depending on the regions/dates?

It would be interesting to have some analysis of DHPS epidemiology depending on the underlying diseases. It could potentially underline the role of prophylaxis, considering the different guidelines for the various underlying diseases.

  • Part 4: Very interesting part and the more controversial subject.

Please consider adding some lines on DHPS mutations impact on the efficacy of TMP SMX as prophylaxis.

Please consider restructuring this part for an easier understanding of the impact of DHPS mutations on 1) TMP SMX prophylaxis efficacy for PCP, 2) PCP curative treatment efficacy: a) severity, b) mortality, c) adverse effect.

Would it be possible that the higher mortality observed in patients with mutations and previous TMP SMX exposure is related to a more severe underlying disease (thus requiring TMP SMW prophylaxis)? Is it what is suggested lines 277-278?

  • Part 5:

Are there any studies reporting DHPS mutations to a genotype of Pneumocystis (in case of MLST)?

Minor comments

Table 1: The table is difficult to read: is there a shift ? Particularly for the column “Amino acid position 55”.

Line 60: I don't think ref 5 (or 6) specifically suggest the possibility of PCP treatment failure?

Table 2: I don’t understand the differences in Portability, discriminator and power and in set up cost between PCR seq and MLST. Doesn’t MLST correspond to DHPS PCR seq+ other genes PCR seq?

Table 2: Would it be possible to add information on the necessary Pj quantity (low, high, very high fungal load/ possible on colonized patients?) for the different methods and the possible detection of subpopulations?

Table 3: Several misspelling: “PCC seq”, “MLS”, MLSP” in the column “DHPS detection”, “no available” in the column period of study.

Line 156: Misspelling MLST

Lines 179-182: Are there any developing countries or regions known for more intensive use of TMP-SMX for PcP or other infections? Conversely, are Asian and Latin American countries known for limited use of TMP-SMX or is it a supposition?

Part 6: line 347-348: Please consider adding references. There is no previous reference to sulfadoxine in the review.

Please consider adding references also lines 43, 55, 86

Lines 355-356: There are probably some words missing in the phrase.

Lines 394-395: please add that it has been tested on other pathogens than Pneumocystis (Saccharomyces and Escherichia).

Author Response

Reviewer 1

Comments and Suggestions for Authors

Thank you for this review focusing on the epidemiology of DHPS mutations in Pneumocystis jirovecii and on their impact on PCP treatment. It gathers a lot of data on this interesting and controversial subject. To my mind, although the different parts are well organized, the structure of the paragraphs composing these parts (especially part 3 and 4) can be a little confusing for it gives a lot of information not necessarly in a logical order (part 4) or gives too many information (part 3).

Thank you for your opinion. We are modified the paragraphs composing in order to clarify.

Major comments

  • Part 2:

Please consider adding some lines on the possibility of mixed infections. It has been reported in several studies, especially using NGS and only evoked lines 86-91 in this review. DOI: 10.3201/eid2308.161295 , DOI: 10.1371/journal.pone.0125763

Thanks. We have included a comment about this issue and added references.

  • Part 3:

Please consider lightening this section of the text, especially as all numbers are already detailed in tables 3 and 4. (discordance between line 199 and study of Beard CB,2000?)

We are happy to do so. There is not discordance. Information in the table is from 5 cities included in the study of Beard. Information in the text is only from San Francisco. However, to evict confusion we have modified the text and included other references.

The relationship between mutations and use of TMP-SMX as prophylaxis is repeated several times in the part 3. Please consider adding references on the use of prophylaxis in the particular cited situations; Or evoking it as a global suggestion concerning the different rate of mutations depending on the regions/dates?

We like this suggestion and have modified the text.

It would be interesting to have some analysis of DHPS epidemiology depending on the underlying diseases. It could potentially underline the role of prophylaxis, considering the different guidelines for the various underlying diseases.

This is an important point. Unfortunately, most of the available information is from HIV patients, therefore it is not possible an analysis based on the underlying diseases.

  • Part 4: Very interesting part and the more controversial subject.

Please consider adding some lines on DHPS mutations impact on the efficacy of TMP SMX as prophylaxis.

We are happy to do so.

Please consider restructuring this part for an easier understanding of the impact of DHPS mutations on 1) TMP SMX prophylaxis efficacy for PCP, 2) PCP curative treatment efficacy: a) severity, b) mortality, c) adverse effect.

This is a good suggestion. We have modified the text in this way.

Would it be possible that the higher mortality observed in patients with mutations and previous TMP SMX exposure is related to a more severe underlying disease (thus requiring TMP SMW prophylaxis)? Is it what is suggested lines 277-278?

It is a possible explanation. The development of mutations could be related to time under TMP-SMX prophylaxis and more time of progression of the underlying disease lead to more severe clinical situation

Part 5:

Are there any studies reporting DHPS mutations to a genotype of Pneumocystis (in case of MLST)?

DHPS is a key genetic region for many MLST studies. Despite its extensive use, low levels of genetic variation have been reported, with most studies reporting wild-type sequences being detected. In fact, new consensus MLST scheme proposed for genotyping do not recommended the inclusion of DHPS locus (Pasic L, et al. J Fungi 2020;6:259)

Minor comments

Table 1: The table is difficult to read: is there a shift ? Particularly for the column “Amino acid position 55”.

Thank you. We have modified it

Line 60: I don't think ref 5 (or 6) specifically suggest the possibility of PCP treatment failure?

We agree. These references directly do not suggest the possibility of PcP treatment failure, suggest chemoprophylaxis failure. However, the fact that DHPS mutations in another microorganisms produce drug resistance leads us to think a similar effect on Pneumocystis. We have added other reference.

Table 2: I don’t understand the differences in Portability, discriminator and power and in set up cost between PCR seq and MLST. Doesn’t MLST correspond to DHPS PCR seq+ other genes PCR seq?

Yes. However, currently there are several Pneumocystis MLST schemes that have been used in 31 published using 19 different loci that including or not DHPS. Their ability to be successfully amplified and sequenced, and their discriminatory power is different among then.

Reproducibility pertains to the ability to obtain the same result after multiple analyses, portability refers to the possibility to exchange data between different laboratories. Discriminatory power of a method is its ability to assign a different type to two unrelated strains sampled randomly from the population of a given species

Table 2: Would it be possible to add information on the necessary Pj quantity (low, high, very high fungal load/ possible on colonized patients?) for the different methods and the possible detection of subpopulations?

Unfortunately, those data are not available for all techniques or are different in different studies

Table 3: Several misspelling: “PCC seq”, “MLS”, MLSP” in the column “DHPS detection”, “no available” in the column period of study.

Thank. We have corrected those misspelling

Line 156: Misspelling MLST

Thank. We have corrected it

Lines 179-182: Are there any developing countries or regions known for more intensive use of TMP-SMX for PcP or other infections? Conversely, are Asian and Latin American countries known for limited use of TMP-SMX or is it a supposition?

The data available about this issue is limit. In the review publishes by de Armas (Parasite, 2011, 18, 219-228) about PcP in developing counties use of chemoprophylaxis in HVI patients was low in the countries where this information was available: 25% in Uganda, 19% in Senegal, 40% in Central African Republic, 7% in Thailand, 4% in Vietnam, 39% in Cambodia or 18% in Chile). We have included this reference.  

Part 6: line 347-348: Please consider adding references. There is no previous reference to sulfadoxine in the review.

We have included a new text and reference

Please consider adding references also lines 43, 55, 86

Thank. We have added references

Lines 355-356: There are probably some words missing in the phrase.

Thank, the phrase has been completed.

Lines 394-395: please add that it has been tested on other pathogens than Pneumocystis (Saccharomyces and Escherichia).

Phrase has been added.

Reviewer 2 Report

This manuscript reviews the knowledge about the mutations in the dihydropteroate synthate (DHPS) of Pneumocystis jirovecii that may confer drug resistance to this fungal pathogen. The review is well written and reads easily, and is fairly complete. I have only few comments.

Major comments

  1. The selection of DHPS mutations by the drug was supported by the analysis of recurrent disease episodes in the same patients (Nahimana EID 2003). This should be added and discussed in the review, and mentioned at line 393.
  2. The actual transmission of mutated P jirovecii strains has been described in a cluster of pneumocystis pneumonia (Rabodonirina EID 2004). This should be stated in this review at lines 327-329.
  3. Lines 315-318: colonized patients as a source of P. jirovecii has been described (Le Gal CID 2012). This should be stated in the review.
  4. lines 57 to 60 : this statment is also strongly supported by the studies having used heterologous expression. This should mentioned here in the review.
  5. Table 2: references should be added within the table rather than in the text. 

Minor comments

  1. line 20: "(DHPS)" should be added after "synthase" because this abbrevaition is used at line 22 in the abstract.
  2. line 22: "and found almost everywhere." is unclear and should be clarified.
  3. line 26 : "new" should be deleted because these aproaches are not new anymore (published several decades ago).
  4. line 197: ref 44 does not seem the correct one.
  5. line 390: replace "as well as"" by "but also in".
  6. Table 2 footnote line 66: the reference Vanhee should be replaced by a reference number and be incorporated in the references list.

Author Response

Reviewer 2

Comments and Suggestions for Authors

This manuscript reviews the knowledge about the mutations in the dihydropteroate synthate (DHPS) of Pneumocystis jirovecii that may confer drug resistance to this fungal pathogen. The review is well written and reads easily, and is fairly complete. I have only few comments.

Thank you for your opinion

Major comments

  • The selection of DHPS mutations by the drug was supported by the analysis of recurrent disease episodes in the same patients (Nahimana EID 2003). This should be added and discussed in the review, and mentioned at line 393.

We are happy to do so. A new paragraph has been included with this information. However, we do not identified this topic in line 393.

  • The actual transmission of mutated P jirovecii strains has been described in a cluster of pneumocystis pneumonia (Rabodonirina EID 2004). This should be stated in this review at lines 327-329.

We agree. The phrase has been included in the text

  • Lines 315-318: colonized patients as a source of P. jirovecii has been described (Le Gal CID 2012). This should be stated in the review.

This information has been included in the text as well as reference

  • lines 57 to 60 : this statment is also strongly supported by the studies having used heterologous expression. This should mentioned here in the review.

We agree. The statement has been included.

  • Table 2: references should be added within the table rather than in the text. 

The reference has been included in the text

Minor comments

  • line 20: "(DHPS)" should be added after "synthase" because this abbrevaition is used at line 22 in the abstract.

Thank. We have modified it.

  • line 22: "and found almost everywhere." is unclear and should be clarified.

The phase has been clarified.

  • line 26 : "new" should be deleted because these aproaches are not new anymore (published several decades ago).

You are right. New has been deleted

  • line 197: ref 44 does not seem the correct one.

We have included other reference

  • line 390: replace "as well as"" by "but also in".

Thank. It is more correct. We have changed it.

  • Table 2 footnote line 66: the reference Vanhee should be replaced by a reference number and be incorporated in the references list.

The reference has been included in the text and replaced by its number in the table
